# Deep Learning Insights into the Dynamic Effects of Photodynamic Therapy on Cancer Cells

**DOI:** 10.3390/pharmaceutics16050673

**Published:** 2024-05-16

**Authors:** Md. Atiqur Rahman, Feihong Yan, Ruiyuan Li, Yu Wang, Lu Huang, Rongcheng Han, Yuqiang Jiang

**Affiliations:** 1State Key Laboratory of Molecular Developmental Biology, Institute of Genetics and Developmental Biology, Chinese Academy of Sciences, Beijing 100101, China; atique@genetics.ac.cn (M.A.R.); yanfeihong@genetics.ac.cn (F.Y.); ryli@genetics.ac.cn (R.L.); yuwang@genetics.ac.cn (Y.W.); huanglu@genetics.ac.cn (L.H.); 2University of Chinese Academy of Sciences, Beijing 100049, China

**Keywords:** photodynamic therapy (PDT), deep learning, instance segmentation, dynamic effects, cancer therapy

## Abstract

Photodynamic therapy (PDT) shows promise in tumor treatment, particularly when combined with nanotechnology. This study examines the impact of deep learning, particularly the Cellpose algorithm, on the comprehension of cancer cell responses to PDT. The Cellpose algorithm enables robust morphological analysis of cancer cells, while logistic growth modelling predicts cellular behavior post-PDT. Rigorous model validation ensures the accuracy of the findings. Cellpose demonstrates significant morphological changes after PDT, affecting cellular proliferation and survival. The reliability of the findings is confirmed by model validation. This deep learning tool enhances our understanding of cancer cell dynamics after PDT. Advanced analytical techniques, such as morphological analysis and growth modeling, provide insights into the effects of PDT on hepatocellular carcinoma (HCC) cells, which could potentially improve cancer treatment efficacy. In summary, the research examines the role of deep learning in optimizing PDT parameters to personalize oncology treatment and improve efficacy.

## 1. Introduction

HCC is the most common form of liver cancer and the sixth most common neoplasm worldwide, resulting in more than 600,000 deaths annually [1,2,3]. HCC arises from an abnormal mass of tumor nodules and spreads to adjacent parts of the liver, ultimately resulting in malignancy [4].

Surgery, chemotherapy, and radiotherapy are common traditional cancer treatments [5]. Although these methods can be effective in treating some cancers and preventing their spread, they also have significant limitations. For instance, surgery is often linked to a high rate of cancer recurrence, while chemotherapy and radiotherapy can cause significant harm to the body. As an alternative treatment, PDT can be considered. This therapy uses light sources to specifically target primary and recurrent cancers, reducing damage to healthy tissue [6]. As a result, PDT is a minimally invasive treatment approach that can enhance patients’ quality of life [7]. PDT has become increasingly popular in various biotechnological fields due to its efficiency, selectivity, and minimally invasive properties [8,9,10].

PDT typically involves the use of a photosensitizer (PS) to transfer energy to molecular oxygen under laser irradiation. This process generates reactive oxygen species (ROS) that ultimately induce cancer cell death [11,12]. To evaluate the therapeutic efficacy of PDT, it is crucial to monitor the dynamic response of cancer cells [13,14].

The development of nanotechnology has also improved PDT. Metal-based nanoparticles, such as manganese dioxide, iron oxide, and cerium oxide, have been increasingly used in PDT to improve therapeutic effects [15,16,17]. In general, ideal PSs for PDT exhibit excellent photodynamic properties while being non-toxic to normal cells in the absence of light irradiation [18,19].

The precise assessment of the effectiveness of PDT has become a critical research focus. It is widely recognized that the state of a cell is closely linked to its morphology. To assess the efficacy of PDT, it is possible to utilize morphological changes that can be observed in bright-field images, obviating the need for fluorescent labelling. However, measuring dynamic changes in cells after PDT using traditional methods can be challenging, particularly for label-free cells. Additionally, quantifying the dynamic process of PDT can be labor-intensive, inefficient, and costly. Therefore, there is an urgent need for developing innovative methods to detect these changes while reducing labor costs, enhancing efficiency, and potentially enabling real-time monitoring.

Deep learning is a powerful method that has been used in biomedical imaging to achieve more reliable and accurate cell instance segmentation and morphological analysis [20]. For instance, the watershed algorithm, which generates a topographical map based on grayscale values with a threshold and divides regions based on pixel intensity, has been employed for cell segmentation [21]. However, the watershed algorithm or its variant is sensitive to noise and inaccurate when cell boundaries are unclear due to cell adhesion. Furthermore, if the size and shape of the cells change, adjustments to factors such as the threshold values are required [22,23]. Therefore, this approach may not be suitable for datasets that require precise segmentation of diverse cell images. As for other advanced methods, such as the U-Net model [24], they can distinguish between backgrounds, cells, and boundaries. However, these approaches are limited to predicting pixels that are part of boundaries in images with high cell density and a high boundary ratio. In addition, large cell boundaries may contain misidentified pixels, causing multiple cells to be misidentified as a single large cell, which can negatively impact performance. Furthermore, models that use star-convex polygons, such as StarDist, may generate multiple polygons for elongated cells, resulting in incomplete cell masks [22,25]. Therefore, a more robust and accurate method for cell instance segmentation is needed, particularly when dealing with diverse cellular morphologies.

Cellpose is a U-net-based convolutional neural network (CNN) with skip connections that uses a style vector to encode the input style. It has been demonstrated to be more resilient to noise than watershed methods and can handle a wider range of cell morphologies. In addition to distinguishing cells and backgrounds per pixel, Cellpose can also be able to predict the gradient of the flow towards the center of each cell, providing richer information than the inference method used in U-Net. Compared to StarDist, Cellpose is more effective at segmenting elongated cells due to its ability to predict the point where the flow gradient of each pixel converges as the center of the cell. These unique properties make Cellpose an ideal choice for cell segmentation tasks in our research, especially for monitoring the effectiveness of PDT.

In this study, Cellpose model was used to evaluate the performance of PDT using L-AuNP@TMT nanoparticles at different laser intensities. After PDT, the treated cancer cells were analyzed to indirectly determine the expression of cell markers in the bright-field images. To segment the diverse morphologies of cancer cells, the Cellpose deep learning algorithm was employed [26]. The use of deep learning in therapeutic evaluation enables the efficient extraction of cell appearances and shapes from bright-field images. Moreover, deep learning models provide significant advantages in assessing cell death, particularly in cancer research, due to their rapid prediction times, which often take only a few seconds. This accelerates the process of monitoring cancer cell viability, resulting in faster and more efficient treatment planning for patients [27]. Deep learning methods have the potential to revolutionize the field of cell death assessment, offering powerful and versatile tools for researchers and clinicians alike. Additionally, their speed is noteworthy [28]. This study aims to contribute to the development of more effective and targeted cancer treatments in the future by providing a comprehensive understanding of the cellular effects of PDT.

## 2. Materials and Methods

### 2.1. Materials

Chloroauric acid hydrated (HAuCl_4_·4H_2_O) was purchased from Shanghai Aladdin Bio-Chem Technology Co., Ltd., Shanghai, China; (5-mercapto-1,3,4-thiadiazol-2-ylthio) acetic acid (TMT) was purchased from Alfa Aesar Co. (Haverhill, MA, USA). Sodium hydroxide (NaOH) was purchased from Beijing Chemical Works (Beijing, China). Dulbecco’s modified Eagle’s medium (DMEM), fetal bovine serum (FBS) and penicillin-streptomycin (P/S) were acquired from Gibco Corporation (Grand Island, NE, USA). All chemicals were used as received without any further purification.

The hepatocellular carcinoma cells (HepG2) were purchased from The National Biomedical Experimental Cell Resource Bank (Beijing, China). The cells were cultured in Dulbecco’s modified Eagle’s medium (DMEM) supplemented with penicillin (100 U/mL), streptomycin (100 mg/L), and 10% fetal bovine serum (FBS) at 37 °C in a humidified atmosphere with 5% CO_2_.

### 2.2. Preparation of L-AuNP@TMT

Luminescent gold nanoparticles, known as L-AuNP@TMT, were synthesized through photocatalytic. The procedure involved the introduction of a 20 mM, 5 mL solution of TMT into a round-bottom flask (Synthware, Beijing, China) containing 39 mL of double-distilled water (ddH_2_O). Subsequently, 1 mL of a 500 mM NaOH solution and 5 mL of a 10 mM gold (III) chloride (HAuCl_4_) solution were added to the mixture. The reaction was carried out for 17 min under continuous, vigorous stirring and 390 nm UV lamp (5 W, self-developed) illumination in a dark environment. Any unwanted by-products were removed via centrifugal filtration at 5500× *g*, using a filter with a 3000 KD cutoff (Merck, Darmstadt, German). This process was repeated six times after the reaction. The gold nanoparticle precipitate was redissolved in ultrapure water (18.2 MΩ.cm, 6 mL) and stored at 4 °C for future use. It is noteworthy that the L-AuNP@TMT nanoparticles exhibited remarkable stability, maintaining their colloidal state free of precipitate for at least three years when stored at 4 °C.

### 2.3. PDT Treatment of HepG2 Cells

HepG2 cells (1 × 10^5^) were cultured on a confocal dish (35 mm) with DMEM culture medium for 24 h at 37 °C and 5% CO_2_. Then, incubated with L-AuNP@TMT (70 μg mL^−1^) at 37 °C for 30 min in the dark. Finally, PDT was performed using an LSM 780 NLO optical system (ZEISS) equipped with a highly versatile and tunable titanium–sapphire (Ti:sapphire) laser, the Mai Tai HP DS. L-AuNP@TMT PDT was performed using a 790 nm femtosecond (fs) laser with varying power levels, carefully and precisely controlled between 6% (4.6 mW) and 12.5% (9.9 mW). The scan area was set at 250 µm × 250 µm, which is suitable for further imaging and analysis. The power density of the laser intensity at 6%, 7%, 8%, 10%, and 12.5% was 7.36 W/cm^2^, 8.64 W/cm^2^, 9.76 W/cm^2^, 12.0 W/cm^2^, and 15.84 W/cm^2^, respectively.

### 2.4. Data Collection and Preprocessing

This study analyzed cellular behavior using fluorescence microscopic time series images to investigate how cells respond to different laser intensities (6% (4.6 mW), 7% (5.4 mW), 8% (6.1 mW), 10% (7.5 mW), and 12.5% (9.9 mW)). A series of time series images were captured for each intensity level. The images were initially in .czi format but were later converted to .jpeg/.png for easier analysis. The dataset comprised 700 images. The images were used to form time series pairs, with 150 pairs from the 6% (4.6 mW) intensity images, 120 pairs from the 7% (5.4 mW) intensity images, 50 pairs from the 8% (6.1 mW) intensity images, and 20 pairs from the 10% (7.5 mW) intensity images. Additionally, 20 individual time points were represented by 20 images captured at 12.5% (9.9 mW) intensity. A total of 16,100 cells were analyzed from 700 images, which were captured at different intensities ranging from 6% (5.4 mW) to 12.5% (9.9 mW). The images had a resolution of 1864 × 1864 pixels.

### 2.5. Deep Learning

A pre-trained Cellpose model (https://github.com/MouseLand/cellpose (accessed on 13 March 2024)) was used in this study. To bypass the need for custom training, we leveraged Cellpose’s extensive pre-training on a vast dataset, achieving accurate instance segmentation for our cells. This approach exploits transfer learning, whereby a pre-trained model’s knowledge is adapted to a novel task (our specific data) through manual correction within the Cellpose framework. This strategy offers several advantages. Firstly, this approach bypasses the time-consuming and resource-intensive process of training a model from scratch. Secondly, the pre-training of Cellpose ensures a robust foundation for accurate segmentation, obviating the necessity for significant manual correction for the delivery of high-quality results. Thirdly, this approach circumvents the intricacies of developing and validating a bespoke model, rendering it accessible and efficacious. Finally, the utilization of a pre-trained model may enhance the generalizability of the model to similar datasets with minimal adjustments, as the core segmentation knowledge has already been established.

Cellpose makes effective use of the power of deep learning architectures, in particular the U-Net architecture [24]. U-Net is renowned for its ability to capture complex details in biomedical images and is the foundation of Cellpose. However, Cellpose employs a modified version of U-Net, further optimized for cell segmentation tasks. This improvement enables Cellpose to efficiently segment cells of various morphologies, sizes, and imaging modalities. A key strength of Cellpose is its versatility in handling diverse cell types and image conditions. In contrast to numerous deep learning models that necessitate extensive retraining for each specific application, Cellpose has been pre-trained on a multitude of cell image collections [26]. This pre-training enables Cellpose to accurately segment cells without the necessity for extensive user intervention, rendering it highly user-friendly and accessible to researchers with varying levels of deep learning expertise.

The characterization of cells involved the extraction of various parameters following segmentation with Cellpose. The quantification of basic dimensional features (length, width, area, perimeter, diameter) and geometric properties (circularity, curvature) of the cells was performed utilizing scikit-image (version 0.23.2), a powerful Python library for image processing. While Excel 2016 was employed for some calculations, scikit-image served as the primary platform for these quantitative assessments.

## 3. Results

### 3.1. Cellular Instance Segmentation Using Cellpose

Figure 1 illustrates the workflow of the deep learning segmentation algorithm for cell image processing, morphological analysis and monitoring the impact of PDT on cancer cells. This workflow provides a comprehensive representation of the segmented elements within the image, which helps in the understanding of the segmentation results and the accuracy of the algorithm. This visual aid is valuable in demonstrating the effectiveness of the instance segmentation method.

Building on the U-Net architecture, which has been tailored for biomedical image segmentation, Cellpose extends this framework by incorporating skip connections. These connections are used to preserve spatial information that is lost during downsampling. This is a crucial aspect of maintaining the integrity of cellular structures. While U-Net primarily distinguishes cells from background cells, Cellpose goes beyond this approach. It provides a more dynamic understanding of cellular structures. This is achieved by predicting the gradient flow towards the center of the cell. In contrast to StarDist, Cellpose excels at handling elongated cells by predicting the point where the flow gradient converges and identifying the center based on the actual shape and orientation of the cell. Additionally, Cellpose employs a style vector to encode the input style, thereby increasing the robustness to variations in cell shape, size and noise level. This adaptability provides a significant advantage over traditional watershed segmentation methods, especially when faced with complex and noisy datasets. This study highlights that Cellpose is emerging as the optimal choice for cell segmentation in research contexts due to its accuracy and versatility across different cell types.

The pre-trained Cellpose model was employed for cell segmentation, incorporating a human-in-the-loop approach [29] to identify and rectify any instances of missing cells. The iterative process of manual correction and retraining was repeated on numerous occasions, resulting in the creation of a bespoke model with enhanced cell prediction accuracy. The ability to accurately segment cells was of paramount importance to the success of this research, and in these cases, Cellpose outperformed other existing cell segmentation models [23]. This approach is of significant importance for further analysis and discussion in our research.

The representative results of the instance segmentation are presented in Figure 2. It is possible to segment cells with different morphological profiles, which allows for further quantitative analysis of the dynamic effect of PDT.

### 3.2. Morphological Changes Induced by PDT

Cellular morphological changes induced by PDT were evaluated with these measurements including length, width, area, perimeter, and diameter, that provide a comprehensive dataset that encapsulates the physical dimensions and overall shape of the cells.

In addition to these basic dimensional measurements, our analysis delved deeper into the geometric properties of the cells by calculating their circularity and curvature, two critical parameters that provide insight into the more nuanced aspects of cell morphology. The circularity of each cell was calculated using the following specific mathematical formula: circularity = (4π × area)/perimeter^2^. This formula is instrumental in quantifying how close the shape of a cell is to a perfect circle, taking into account its area and perimeter. A higher circularity indicates a shape closer to a circle, which can be an important indicator in various biological and medical analyses.

We also assessed the curvature of the cells, a measure that gives an indication of the shape of the cell in terms of its curvature or arching. Curvature was calculated based on the radius, which was calculated as the average of the cell length and width, specifically using the following formula: radius = (length + width)/4. Curvature, defined as the reciprocal of the radius (i.e., curvature = 1/radius), reflects the degree of curvature of the cell outline and provides further insight into the dimensional structure of the cell. Such detailed morphological analysis is crucial in several scientific and medical fields, as variations in cell size, shape and structure are often key indicators of physiological or pathological states.

Using Cellpose, a total of 16,100 cells were segmented in 700 time series images, revealing significant morphological changes induced by PDT. The effect of different laser intensities was evident in the observed changes in cell size, shape, and overall appearance.

### 3.3. Analysis the Impact of PDT on Cancer Cells

The size and shape of the cells were examined. Larger cells are bigger and more curved, which facilitates the comprehension of cell behaviors and properties. Figure 3 represents the mean values of cell area, curvature, and circularity. These features facilitate the understanding and comparison of the cells in our study. The images in Figure 3 demonstrate how cells can change in different ways, which assists in the comprehension of cell functioning and the activities they engage in. The left graph (Figure 3A) presents a timeline with a more detailed scale, while the right graph (Figure 3B) appears to be a magnified version showing trends over a longer timeframe. The changes in the cellular features over time at different laser intensities are represented by lines of varying colors. These graphs demonstrate how these features changed under different conditions, thereby facilitating an understanding of how cells behave in response to different laser intensities over time. The graph illustrates the interrelationship between time, laser intensity and cell shape. As time progresses and the laser intensity increases, the cells undergo a reduction in size and an increase in roundness.

As a result, we observed that noticeable changes in cell morphology over time progressed and the laser intensity increased. Specifically, the cells underwent a process of shrinkage and assumed a more circular shape. This finding implies that PDT is successful in targeting cancer cells, and that its effectiveness can be improved by modifying two important factors: the intensity and duration of laser exposure. The application of a more intense laser intensity results in an augmented production of ROS, and an excess of ROS is deleterious to the essential elements within cancer cells, such as lipids. Consequently, this process causes cancer cells to shrink and take on a circular shape. This observation indicated that there was a reduction in the average size of the cells. Simultaneously, there was an increase in both the average curvature and the circularity of the cells.

### 3.4. Model Validation

To evaluate the model, we used a dataset where each image contained 24 cells and was captured with a laser intensity of 4.6 mW. Established statistical metrics—root mean square (RMS), exponential mean absolute error (EMAE), mean squared error (MSE), mean absolute percentage error (MAPE), and coefficient of determination (R^2^)—were utilized to evaluate Cellpose’s accuracy. A comprehensive analysis of cell image segmentation and prediction accuracy is presented in Figure 4.

The statistical results were highly encouraging and showed a remarkable degree of accuracy. The Cellpose model achieved an impressive score of approximately 86.43%. Furthermore, a low MAPE value of 0.1357 and an R^2^ value close to 0.84 further corroborated the model’s reliability for cell size measurement tasks. These findings suggest that Cellpose can effectively and precisely quantify cell size across a range of image datasets.

Nevertheless, it is of vital importance to consider alternative deep learning models that are commonly employed for cell instance segmentation, such as StarDist. While StarDist is renowned for achieving exceptional accuracy in cell segmentation, particularly for densely packed or overlapping cells, it possesses a significant limitation. StarDist is primarily designed for star-convex or rounded cell shapes (more in the Appendix A). This can be a disadvantage in scenarios where the cell population exhibits a high degree of morphological diversity.

In contrast, Cellpose demonstrates a distinct advantage—its ability to handle a wider range of cell shapes. This versatility makes Cellpose a compelling choice for research involving heterogeneous cell populations, where cells may not conform to a single, well-defined morphology.

### 3.5. Logistic Growth Model Analysis

The logistic sigmoidal growth model applied to the average circularity values provided insight into the saturation effects of PDT. Figure 5 shows the fitted logistic growth curves against time for different laser intensities. The parameters of the logistic growth model are detailed in Table 1.

The formula is given as follows: f(t) = A_2_ + (A_1_ − A_2_)/(1 + exp((t − x_0_)/dx)), where f(t) is the value of the function at time t; A_1_ is the initial value of the function, representing the lower asymptote; A_2_ is the upper asymptote or the curve’s maximum value; exp is the exponential function; t is the time or the independent variable; x_0_ is the value of t at the midpoint of the sigmoid curve; dx is a scaling parameter that determines the steepness of the curve.

The key parameters of this model, such as ‘A_2_’ (the upper asymptote) and ‘x_0_’ (the midpoint of the sigmoid curve), are critical to understanding the growth dynamics of cellular responses to PDT. Analysis of these parameters, detailed in Table 1, provides insight into optimizing PDT protocols to ensure the most effective use of laser intensity over time for inducing the desired cellular changes.

In the context of average circularity values, the application of this function could help to understand how certain shapes or structures develop and grow over time, particularly if they exhibit growth patterns that start slowly, accelerate and then stabilize. This type of modeling is crucial in fields such as biology, where it can be used to understand tumor growth, the expansion of bacterial colonies or the spread of disease, as well as in technology and social sciences to model the adoption rates of innovations or population dynamics.

The increase in average circularity suggested that the cells adopted a more rounded shape. This observation led to the application of a logistic growth model to the data, a model often used to describe saturation effects such as population growth reaching a limit. Here, the impact of PDT on cancer cells was analyzed. The model aims to identify the saturation point at which PDT becomes ineffective.

The logistic sigmoidal growth model, when applied to the relative circularity data presented in Figure 5, offers a quantitative understanding of the impact of different laser intensities on cell morphology during PDT. This approach demonstrates a clear correlation between the treatment variables (laser intensities) and the cellular responses observed in terms of average circularity. The sigmoidal curve indicates an initial phase of minimal change in circularity, followed by a steeper decline as the laser intensity increases. This trend indicates a progressive loss of cell roundness, which may suggest a transition towards cell death at higher intensities. The quantification of these morphological alterations provides valuable insights into the effectiveness of PDT at different treatment settings.

## 4. Discussion

Accurately measuring the onset time of anti-cancer drugs for different drug administration methods is indeed a major challenge in cancer research. Traditional methods, such as creating time gradients in in vitro experiments, are commonly used [30]. However, these techniques are highly dependent on the skill and experience of the operator, which can lead to problems with repeatability and potential bias in the results [31]. This reliance on manual expertise can result in variability in experimental results, a concern that has been raised [32]. In addition, studies have advocated the use of more automated and objective methods to reduce bias and increase measurement reliability [33]. These findings highlight the complexity of measuring the onset times of anticancer drugs and the need for more robust, reproducible methodologies in the field of cancer pharmacology research. In this study, the use of Cellpose for the cellular instance segmentation and analysis of cellular instances following PDT on HCC cells showed remarkable results, contributing significantly to the current landscape of cancer research and treatment methodologies. Our findings are in line with the pioneering work on Cellpose, demonstrating its superior accuracy and adaptability in handling different cellular morphologies, a key advancement compared to traditional segmentation methods [26].

Upon completion of the instance segmentation phase, we embarked on an extensive measurement and analysis of various morphological attributes of all cells within the segmented images, utilizing the robust functionality of Python library. These measurements included length, width, area, perimeter, and diameter, providing a comprehensive dataset that encapsulated the physical dimensions and overall shape of the cells. In addition to these basic dimensional measurements, our analysis delved deeper into the geometric properties of the cells by calculating their circularity and curvature, two critical parameters that provide insight into the more nuanced aspects of cell morphology. A higher circularity indicates a shape closer to a circle, which can be an important indicator in various biological and medical analyses. Curvature, defined as the reciprocal of the radius (i.e., curvature = 1/radius), reflects the degree of curvature of the cell outline and provides further insight into the dimensional structure of the cell. Such detailed morphological analysis is crucial in several scientific and medical fields, as variations in cell size, shape and structure are often key indicators of physiological or pathological states. Finally, the application of logistic sigmoidal growth modeling further enhanced our understanding of how certain cellular structures develop and grow over time and laser intensities. As summarized in Figure 5, the intricate relationship between circularity values and the dynamics of PDT-induced morphological changes was elucidated.

Our approach is consistent with the methodologies utilized in studies [34,35], which emphasize the importance of detailed cellular analysis in understanding the efficacy of cancer treatments. Our study extends this understanding by offering insights into the specific morphological changes induced by PDT, highlighting the treatment’s impact on cell size and shape.

Furthermore, the application of logistic sigmoidal growth modeling in our research to analyze the saturation effects of PDT is consistent with the growing need for more robust and reproducible methodologies in cancer pharmacology research. This approach, which provides insight into the growth dynamics and saturation points of cellular responses to treatment, is particularly relevant given the challenges in accurately measuring the onset times of anti-cancer drugs, as highlighted in [36]. By employing this model, our study provides a nuanced understanding of how PDT affects cancer cells over time, contributing valuable data for optimizing treatment protocols.

The high accuracy of the deep learning model, validated through various statistical metrics, underscores its reliability and effectiveness in measuring and predicting cellular morphology and responses. This aspect of our research resonates with the call for more automated and objective methods in cancer research, a sentiment echoed by [37]. The validation of our model not only confirms its efficacy but also suggests the potential of deep learning methods to improve the precision of cancer cell analysis, thereby aiding in the development of more effective cancer therapies.

## 5. Conclusions

In conclusion, this study demonstrates the potential of deep learning to deepen the understanding of PDT and its dynamic effects on cancer cells. The use of Cellpose, an advanced instance segmentation algorithm, provides unprecedented insight into the complex cellular responses induced by PDT. Deep learning provides a powerful tool for extracting nuanced patterns from high-dimensional datasets and unraveling the complex dynamics of PDT, overcoming the limitations of traditional quantification methods. This innovative approach holds immense promise for enhancing the understanding of PDT mechanisms and optimizing treatment protocols. The integration of deep learning into PDT research represents a significant step forward in advancing our understanding of cancer treatment modalities. Accurately monitoring and predicting the dynamic effects of PDT on individual cancer cells has the potential to enable personalized therapeutic strategies. Additionally, we aim to explore the use of novel deep learning and artificial intelligence models to enhance the efficiency and accuracy of PDT assessment. To effectively evaluate the therapeutic effects of nanophotosensitizers and contribute to the advancement of nanomedicine, we plan to further develop new methods to assess a wider range of tumor cells in the future.

## Figures and Tables

**Figure 1 pharmaceutics-16-00673-f001:**
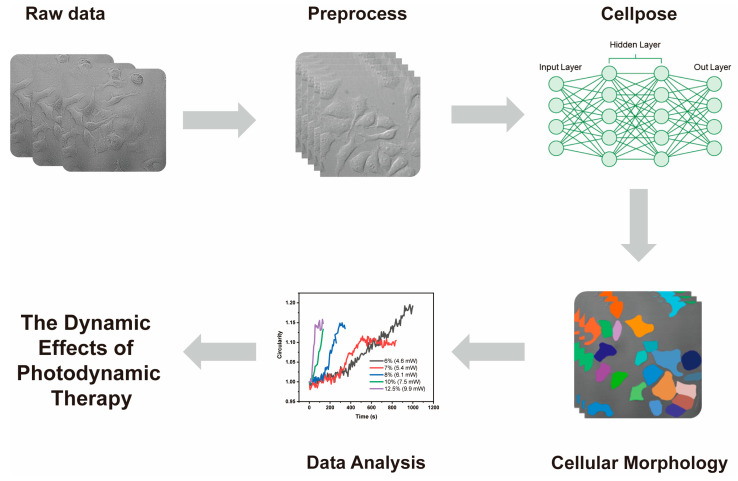
Schematic of the workflow of the deep learning segmentation algorithm for cell image processing, morphological analysis and monitoring the impact of PDT on cancer cells.

**Figure 2 pharmaceutics-16-00673-f002:**
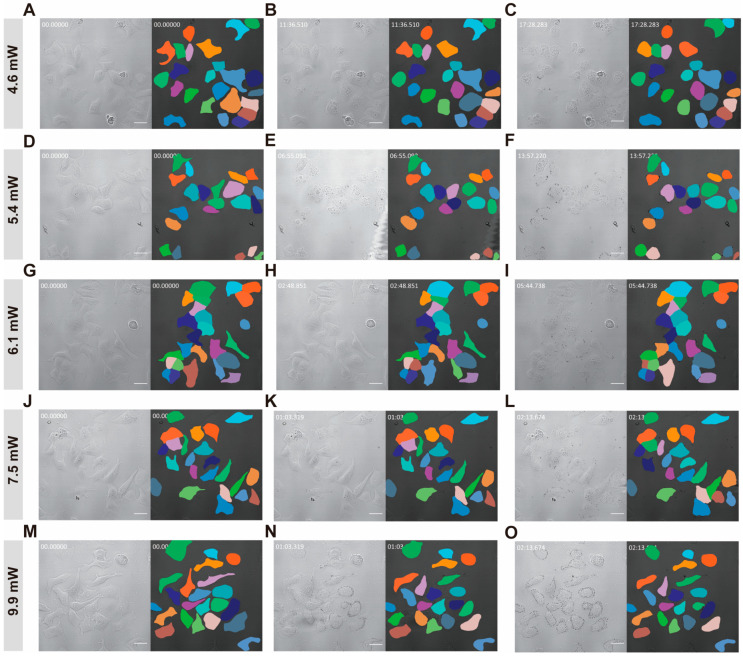
Typical instance segmentation results obtained by the Cellpose algorithm at different time points and laser intensities ranging from 6% (4.6 mW) to 12.5% (9.9 mW). The white lettering in the upper left corner of the image represents the relative time point, with a time format of mm:ss.sss: (**A**–**C**) 4.6 mW, (**D**–**F**) 5.4 mW, (**G**–**I**) 6.1 mW, (**J**–**L**) 7.5 mW and (**M**–**O**) 9.9 mW, respectively. Scale bars: 30 μm.

**Figure 3 pharmaceutics-16-00673-f003:**
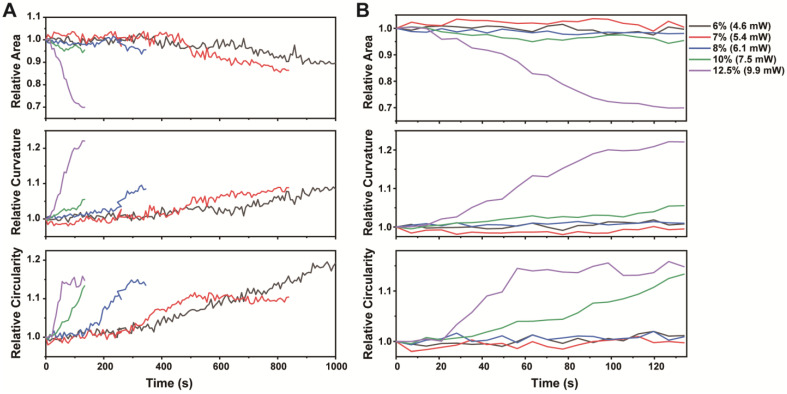
Graphs (**A**,**B**) illustrate changes in the averaged area, curvature, and circularity of cells over time, influenced by varying laser intensities. Graphs (**A**) shows the entire time series images, while (**B**) focuses on the first 20 images in the series. The laser intensities are indicated by distinct colored lines in each graph.

**Figure 4 pharmaceutics-16-00673-f004:**
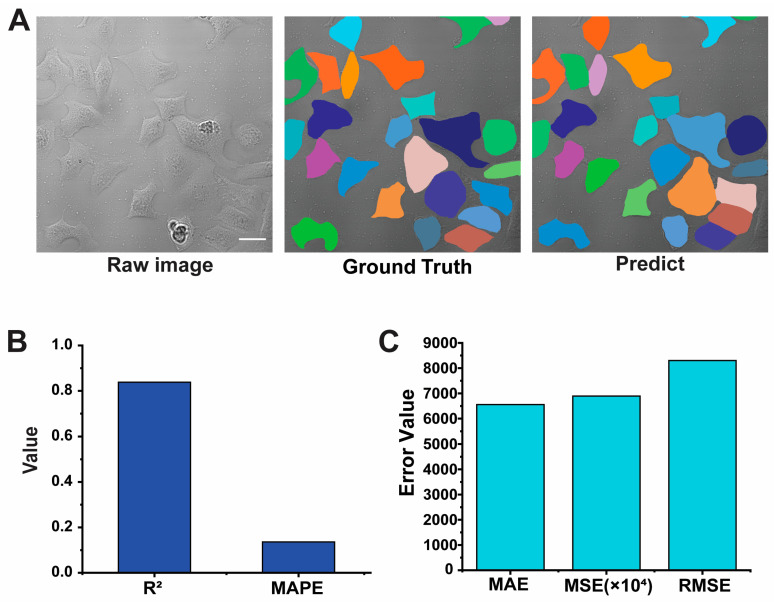
The results of a three-part analysis of cell imaging and prediction accuracy. (**A**) The raw cell image, the precise ground truth cell segmentation, and the Cellpose model’s cell segmentation prediction. Scale bar: 30 μm. (**B**,**C**) correspond to a bar chart comparing the mean absolute percentage error (MAPE) and the coefficient of determination (R^2^) values, and a bar chart with scaled mean absolute error (MAE), mean squared error (MSE) and root mean squared error (RMSE) values for model prediction performance. Furthermore, the MAE, MSE, and RMSE values provide additional insights into the model’s accuracy.

**Figure 5 pharmaceutics-16-00673-f005:**
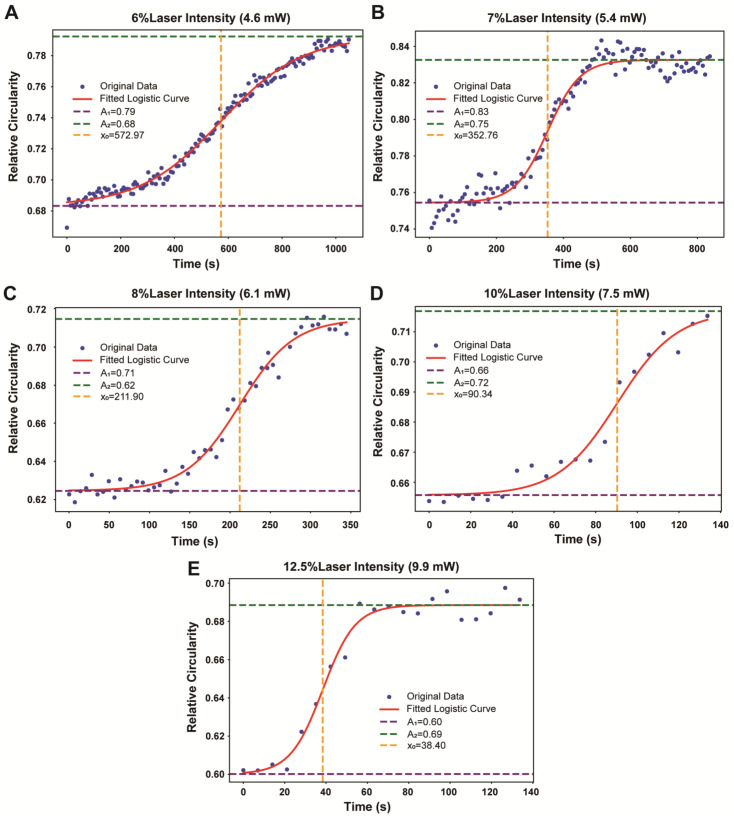
Typical dynamic profiles of cellular circularity in different groups with different PDT laser intensities. The data points, represented by blue dots, represent the observed experimental measurements of cell circularity at different time intervals. The orange curve represents the fitted logistic growth model superimposed on these data points. This model is characterized by an ‘S-shaped’ growth pattern: an initial slow growth phase, followed by a rapid exponential increase and a final plateau at a maximum limit. The results obtained under laser intensities of 4.6 mW, 5.4 mW, 6.1 mW, 7.5 mW, and 9.9 mW, respectively, are as follows: (**A**–**E**).

**Table 1 pharmaceutics-16-00673-t001:** The parameters of the logic model related to morphological profiles in the cells, including area, perimeter and aspect ratio at different laser intensities. The dynamic effect of PDT can be evaluated by X_0_, which indicates the laser intensity at which the feature reaches half the difference between A_1_ and A_2_, with smaller values indicating greater cell harm and more pronounced morphological changes.

Parameters/Laser Intensity	6% Laser Intensity (4.6 mW)	7% Laser Intensity (5.4 mW)	8% Laser Intensity (6.1 mW)	10% Laser Intensity (7.5 mW)	12.50% Laser Intensity (9.9 mW)
A1	0.68	0.75	0.62	0.66	0.60
A2	0.79	0.83	0.71	0.72	0.69
X0	572.97	352.76	211.90	90.34	38.40
dx	150.57	48.40	33.54	14.27	7.84

## Data Availability

Data are contained within the article and Appendix A.

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
