# Peer review of "Deep Learning Insights into the Dynamic Effects of Photodynamic Therapy on Cancer Cells"

_pharmaceutics, 2024, doi:10.3390/pharmaceutics16050673_

Round 1

Reviewer 1 Report

Comments and Suggestions for Authors

The authors present a study on the use of deep learning to understand photodynamic therapy on liver cell cancer cells.

Although the subject is original and exciting, the article is much less so.

There's a lot of confusion between the material and method, results and discussion paragraphs. Many paragraphs are not in the right place. The entire text has to be rewritten

What's more, the results are badly commented on, which is a shame.

Some remarks:

P1 line 6: The authors said that PDT is a non-invasive therapy, which is true for tumours that are easily accessible to a laser fibre, but for deep tumours such as hepatocellular carcinoma, in situ treatment requires access to the tumour, which is invasive? Can the authors qualify their statement?

P3 line 120: Can the authors explain why they used a UV lamp at 390nm to prepare their photosensitizer?

Page 4 lines 166 - 182: paragraph not useful, to be removed.

Page 4 lines1 84-186 : paragraph not useful, to be removed.

Page 5 line 193-195: already said before, sentence to be removed : « cellpose… stuctures »

Pages 5 -12: the entire materials and methods section needs to be rewritten, as it is mixed up with the discussion and even the introduction.

Page Fig 2 : can the authors add the scale?

Page 7 lines 256-259 : already said before, sentence to be removed

Page 8 figure 3: what are the units of each Y axis?

Why is the curve corresponding to 12.5% irradiation in yellow and not purple in Figure B?

Page 9 line 233: Did the authors measure the ROS produced in their experiments?

Page 10 figure 4: can the authors add the scale? For greater clarity, could the authors explain the acronyms used?

Page 11 table 1: Could the authors make the table legend more explicit? For example, indicate which morphological changes have been taken into account, indicate what corresponds to A1, A2, Xo and dx. Readers should be able to understand quickly without reading the text (even if it says so just before).

Page 12 Figure 5: For each graph, can the authors indicate the laser intensity used?

Page 12 lines 400-403: Can the authors develop this paragraph?

Page 13 line 443: “to revolutionize”, excessive, please reword this sentence

Out of curiosity, did the authors observe a bystander effect on non-illuminated cells adjacent to those illuminated by the laser?

Page 13 lines 454-455: redundant with what has already been said above, to be removed

Reviewer 2 Report

Comments and Suggestions for Authors

The article entitled: “Deep Learning Insights into the Dynamic Effects of Photody- 2 namic Therapy on Cancer Cells” is potentially a landmark in the field of diagnostic.

As far as this impressive technology is considered, some remarks would improve manuscript soundness.

Irrespective of the HCC cells used in this validation model, broader types of neoplastic cells and different photosensitizers testing in a near future may further enhance deep learning as a diagnostic tool.

However, the PDT treatment of the HepG2 cell is poorly described in the manuscript. Authors are encouraged to describe: tip diameter, distance of the tip to the culture, irradiation time, fluence and total energy delivery of the different “laser intensities” used, as scientific recommended by WALT (World Association for photobiomodulatory therapy)

Reviewer 3 Report

Comments and Suggestions for Authors

This paper investigates the impact of deep learning, specifically the Cellpose algorithm, on understanding cancer cell responses to photodynamic therapy (PDT) in the context of tumor treatment. The study aims to enhance comprehension of PDT's efficacy against cancer, employing advanced analytical techniques such as morphological analysis and growth modeling. The main contributions lie in the application of deep learning to elucidate cellular dynamics post-PDT and the potential for optimizing treatment parameters for personalized oncology. The paper presents a compelling concept that integrates cutting-edge technologies to address an important aspect of cancer treatment. By leveraging deep learning and sophisticated analytical methods, the study offers a novel approach to understanding the mechanisms underlying PDT's effectiveness against cancer cells. The exploration of PDT in conjunction with nanotechnology further enriches the discussion, highlighting the multidisciplinary nature of the research. The review topic is comprehensive, encompassing the influence of deep learning on understanding cancer cell reactions to PDT. The authors effectively identify the gap in knowledge regarding the application of advanced analytical techniques in this context, thereby addressing a pertinent issue in cancer research. Table and figures are appropriate and easy to interpret.

Round 2

Reviewer 1 Report

Comments and Suggestions for Authors

This version of the article is much better, and the authors have done an excellent job of clarity and presentation.

It's very good. Bravo